# Reduced Nitrogen Input Combined with Nitrogen-Saving *japonica* Rice Varieties Balances Yield and Nitrogen Use Efficiency in The Lower Reaches of the Yangtze River in China

Xiaoxiang Zhang [1,2], Honggen Zhang [1], Zi Wang [1], Yingbo Gao [1], Xin Liu [1], Xiaowei Shu [1], Yueqi Chen [1], Ning Xiao [2], Cunhong Pan [2], Juan Zhou [1], Chunming Ji [2], Guichun Dong [1], Niansheng Huang [2], Jianye Huang [1], Aihong Li [2,*] and Youli Yao [1,*]

[1] Jiangsu Key Laboratory of Crop Genetics and Physiology/Co-Innovation Center for Modern Production Technology of Grain Crops, Yangzhou University, Yangzhou 225009, China; zhngyz@126.com (X.Z.); zhg@yzu.edu.cn (H.Z.); w1564473824@163.com (Z.W.); yingbogao_yzu@163.com (Y.G.); liuxinkiq@163.com (X.L.); 18762304072@163.com (X.S.); cyq1042844164@163.com (Y.C.); juanzhou@yzu.edu.cn (J.Z.); gcdong@yzu.edu.cn (G.D.); jyhuang@yzu.edu.cn (J.H.)
[2] Lixiahe Agricultural Research Institute of Jiangsu Province, Yangzhou 225007, China; xn_yzu@126.com (N.X.); pancunhong@163.com (C.P.); jcmcn@163.com (C.J.); jsyzhns@163.com (N.H.)
[*] Correspondence: ahli@163.com (A.L.); yaoyl@yzu.edu.cn (Y.Y.)

**Abstract:** Maintaining rice yield and reducing nitrogen (N) input are two important targets in sustainable agriculture practices. The adoption of a nitrogen-saving variety (NSV) provides a unique opportunity to achieve this. However, limited options in NSV *japonica* rice and a lack of information on their responses to N reduction make management decisions difficult. This study aims to explore the responses of yield and nitrogen use efficiency (NUE) in NSV to N reduction. Two newly released NSVs and two popular general varieties (GVs) of *japonica* rice were field tested in Yangzhou, located at the lower reaches of Yangtze River of China, in two consecutive years. The results showed that for NSVs, with a 40–60% reduction in common practice N rate (300 Kg ha$^{-1}$), the rice yield could maintain a record average level ($p < 0.05$), whereas the yield for the GV would drop 20–30% ($p < 0.05$). This indicates that combining the practices of adoption of NSV and N reduction to 120–180 Kg N ha$^{-1}$ could balance the yield and N consumption. Moderate N reduction promotes the N accumulation and NUE, and it increases the number of tillers, the productive tiller percentage and the total amount of spikelets in the population, and increases the carbon and N metabolism of the population in the NSV. Compared with GV, NSV showed higher NUE and non-structural carbohydrate re-mobilization in the reduced N rate. The results showed that the practice of N reduction has to adopt NSV at the same time in order to maintain the grain yield level in rice.

**Keywords:** nitrogen-saving *japonica* rice (*Oryza sativa* L.); nitrogen rate; grain yield; nitrogen use efficiency; non-structural carbohydrate

## 1. Introduction

China is the world's largest producer and consumer of rice, and more than 60 percent of its population relies on rice as a staple food [1–3]. Therefore, rice production plays a central role in ensuring its national food security in China [4,5]. Nitrogen (N) is the most important nutrient for rice production [1,6]. N fertilization is the most significant management approach in the regulation of rice growth, development, and yield formation [7,8]. A reasonable N fertilizer management can optimize the rice population structure, reduce ineffective tillers, maintain strong photosynthetic activity and high non-structural carbohydrate (NSC) transport efficiency in the later growth stage, and, thus, increase the rice yield [9,10]. Maintaining a certain paddy yield level arguably depends on the input of N fertilizer [11,12], considering that the cropping acreage is generally plateaued. Located at the lower reaches of the Yangtze River, Jiangsu Province is a major rice producer,

and its paddy yield ranks top in China [1,13]. These proud ranks come out of an agonizing N input in the region, especially for the conventional *japonica* rice varieties, reaching 300 kg ha$^{-1}$, which is 50% more than the national average [14,15]. Recent national and provincial policy regulations of N input have successfully shifted the N application rate down to 240–255 kg ha$^{-1}$, without a significant sacrifice in terms of the paddy yield, indicating that a balance of yield and N use efficiency can be achieved [16]. However, the current N input in Jiangsu Province is still a much greater rate than its neighboring regions and countries of similar latitude, causing many issues, such as an overly high rice production cost, low N use efficiency, and environmental sacrifices, including soaring carbon footprint and eutrophication [17].

The other half of the story lies in the shift to the adoption of high-yielding *japonica* rice varieties in Jiangsu Province since the middle of the 1990s [18]. Many reports point out that these erect panicle *japonica* varieties rely on excessive N input to achieve the target yield [19]. Recent research and breeding efforts have revealed that complex processes are involved in N use efficiency of a variety [20]. However, there are few clarifications of the yield and N use efficiency levels of the recently released *japonica* rice varieties in the province, especially under a reduced N rate. Based on preliminary trials, the responses of two newly released N-saving varieties were compared with existing popular *japonica* varieties. The novelty of this study was the discovery of new rice varieties that can still obtain high yields under low-N conditions. The utilization of N-saving varieties (NSVs) is a good breakthrough to balance the yield and fertilizer input. The hypothesis was that further reductions in N input must be combined with a proper N-saving variety in order to achieve a balance of paddy yield and N rate. The results can provide a valuable reference for breeding N-saving rice varieties and managing N reduction practices in sustainable production.

## 2. Materials and Methods

### 2.1. Field Conditions and Rice Varieties

Field experiments were performed in a farmer's field in Jiudian (32°29′ N, 119°26′ E), Hanjiang of Yangzhou Municipal City, Jiangsu Province of China, in two consecutive years (2021 and 2022). The soil was sandy loam type with a medium soil fertility, containing organic matter 25.76 g kg$^{-1}$, alkali hydrolysable N 1.20 g kg$^{-1}$, Olsen phosphorus (P$_2$O$_5$) 45.81 mg kg$^{-1}$, and exchangeable potassium (K$_2$O) 82.32 mg kg$^{-1}$. The previous wheat crop produced grain yield of 7.21 t ha$^{-1}$, and the wheat straw was fully returned to the soil. The weather conditions, including average temperature, solar radiation, and precipitation, during the rice-growing season are shown in Table 1.

**Table 1.** Average temperature, solar radiation, and precipitation in rice-growing season.

| Month | Mean Temperature (C°) | | Solar Radiation (MJ m$^{-2}$ per Month) | | Precipitation (mm per Month) | |
|---|---|---|---|---|---|---|
| | 2021 | 2022 | 2021 | 2022 | 2021 | 2022 |
| May | 22.3 | 21.5 | 391 | 320 | 53.9 | 12.3 |
| June | 26.5 | 28.2 | 513 | 521 | 17.4 | 35.4 |
| July | 28.5 | 30.1 | 501 | 523 | 143.8 | 27.5 |
| August | 28.1 | 30.5 | 532 | 485 | 46.1 | 32.5 |
| September | 26.7 | 23.5 | 387 | 367 | 7.5 | 3.3 |
| October | 19.3 | 17.5 | 369 | 348 | 88.2 | 67.6 |

Mean temperature value was the monthly average. Solar radiation value was monthly totals. Precipitation value was monthly totals.

Four *japonica* rice (*Oryza sativa* L.) varieties, including two recently released N-saving varieties (NSVs), Wumijing (WMJ) and Yangnongjing (YNJ), which can obtain high yield under low-N conditions, and two popular general rice varieties (GVs), Huaidao 5 (HD5) and Mugengjing 200 (MGJ200), were provided by Lixiahe Agriculture Research Institute. All these varieties belong to late-type mid-season-maturity *japonica* rice varieties, with a growth period of 147–150 days in the lower reaches of Yangtze river in China. Per personal

communication, the breeders of the NSV claim they respond to N rate in a very sensitive manner; therefore, a comparison experiment was scheduled.

### 2.2. Experimental Design and Growth Conditions

The seedlings were prepared according to the local machine-transplanting nursery method. The rice seeds were sterilized, imbibed, and enhanced under 32 °C for 12 h before being sown in muddy nursery bed on 25 May and transplanted on 15 June. The experiments adopted a split-plot design, with N rate as the main plot and variety as split plot. The plot area was 36 m$^2$ with three replications. Four seedlings were transplanted in each hill, at a spacing of 30.0 cm $\times$ 10.60 cm (31 hills m$^{-2}$, 2.1 $\times$ 10$^4$ hills·666.67 m$^{-2}$).

Five rates of total N fertilization were set up, N0, N8, N12, N16, and N20, representing 0, 8, 12, 16, and 20 kg N·666.67 m$^{-2}$, corresponding to 0, 120, 180, 240, and 300 kg N·ha$^{-1}$, respectively. The local common N fertilization rate was N20. The N fertilizer (urea, 46.67% N) was split as basal, tiller promotion, and spikelet promotion at a ratio of 3:4:3, respectively (Table 2). The basal fertilizer was top-dressed 1 day before transplanting, the tiller promotion fertilizer further two-way split at 7 and 14 days after transplanting, and spikelet promotion fertilizer at the emergence stage of the third leaf top-down (around spikelet differentiation stage). The phosphorus (120 kg ha$^{-1}$ as P$_2$O$_5$) and potassium (120 kg ha$^{-1}$ as KCl) fertilizers were applied as basal fertilizers one day before transplanting. Field management of water, weeds, and pests followed local conventional practices, as described previously [19].

**Table 2.** The amount and application stage of reduced nitrogen fertilizer used in this study.

| Nitrogen Rate | Total Nitrogen (kg ha$^{-1}$) | Base Fertilizer (kg ha$^{-1}$) | Tiller Fertilizer (kg ha$^{-1}$) | Spikelet-Promoting Fertilizer (kg ha$^{-1}$) | Ratio |
|---|---|---|---|---|---|
| N20 | 300 | 90 | 120 | 90 | 3:4:3 |
| N16 | 240 | 72 | 96 | 72 | 3:4:3 |
| N12 | 180 | 54 | 72 | 54 | 3:4:3 |
| N8 | 120 | 36 | 48 | 36 | 3:4:3 |
| N0 | 0 | 0 | 0 | 0 | / |

N20: no nitrogen reduction, conventional nitrogen fertilizer treatment; N16: 20% reduction in conventional nitrogen rate; N12: 40% reduction in conventional nitrogen rate; N8: 60% reduction in conventional nitrogen rate; N0: zero nitrogen fertilizer application.

### 2.3. Sampling and Measurements

Tiller and population monitoring was determined by enumerating stem and tillers from 30 marked hills from each plot, at a seven-day interval until heading. Five representative hills of plants were sampled to determine the biomass accumulation at the middle stage of tillering, initial stage of panicle differentiation, heading, and maturity stages, according to the average number of stems and tillers in the plot. The shoot part of the rice plants was separated into stems, leaves, and panicles (after heading), then subjected to drying at 105 °C for 30 min and 82 °C to constant weight before being weighed.

### 2.4. Determination of Yield and Its Components

At maturity stage, 50 hills were surveyed in each plot to determine the number of productive panicles. Fifteen representative hills of plants were harvested to determine the number of panicles, spikelet number per panicle, seed setting rate, and 1000-kernel weight, selected according to the average number of productive panicles. One square meter was harvested from each plot to determine paddy grain yield, and the grain water content was adjusted to 14.5% for analysis.

*2.5. Determination of Total N Content, Various N Use Efficiencies, and Estimation of Apparent Remobilized Non-Structural Carbohydrates (NSCs)*

After grinding the collected biomass, 0.5 g powder from each sample was digested for N content determination, following Kjeldahl N analyzer protocol (FOSS Kijeltec 8400). N uptake and utilization efficiencies were calculated according to the following methods:

- Remobilized NSC in stem and leaves (Kg) = amount of biomass in stem and leaves at heading—amount of biomass transfer in stem and leaves at maturity;
- Ratio of remobilization of NSC in stem and leaves (%) = remobilized NSC in stem and leaves/amount of biomass in stem and leaves at heading;
- N accumulation (NA, kg ha$^{-1}$) = dry matter weight in the above—ground $\times$ N content;
- N use efficiency for grain (NUEg, kg grain kg$^{-1}$ N) = grain yield/N accumulation at maturity;
- Agronomic N use efficiency (NAE, kg grain kg$^{-1}$ N) = (grain yield in N application plot—grain yield in N blank plot)/N rate;
- N partial factor productivity (NPFP, kg grain kg$^{-1}$ N) = grain yield/N rate;
- N physiological efficiency (NPE, kg grain kg$^{-1}$ N) = (grain yield in N application plot—grain yield in N blank plot)/(N accumulation in N application plot—N accumulation in N blank plot);
- NSC contribution to grain (%) = (NSC content in stems and sheaths at heading stage—NSC in stems and sheaths at maturity stage)/total grain yield.

*2.6. Statistical Analysis*

Analysis of variance (ANOVA) and comparison of treatments were performed using the General Linear Model procedure and least significant difference test (LSD, $p = 0.05$) from IBM SPSS (IBM SPSS 25.0, Inc., Chicago, IL, USA). Correlation analysis was carried out using Pearson correlation and path analysis using correlation procedures. The graphs are presented using Origin 2022 (Origin Lab, Hampton, MA, USA).

**3. Results**

*3.1. Grain Yield and Yield Components*

ANOVA results indicated that the effects of year and N rate and their interactions were significant to grain yield (Tables 3 and S1). The yield's response to reduced N rate was similar in the two years.

The paddy yield outputs from the local common N fertilization rate (N20) of the four varieties ranged from 9.3 to 10.3 t ha$^{-1}$ (Tables 3 and S1), comparable to the high-end records of the rice yield from the region. These indicated that the experimental field conditions were representing the local management practices. Under reduced N treatment (N8, N12 and N16), the difference in the grain yield between the N-saving varieties (NSVs) and general varieties (GVs) was significant ($p < 0.05$), and the yield of NSVs was significantly greater than that of GVs ($p < 0.05$) (Figure 1a,b). The NSV produced the highest yield at N12, significantly higher than those of N8 and N16, mainly contributed by the increased number of panicles and spikelets per panicle (Tables 3 and S1). The NSV Yangnongjing (YNJ) consistently produced the highest grain yield in the two seasons, whereas the GV Huaidao5 (HD5) had the lowest ones.

However, the response curve of the yield to N rate displayed a different pattern between the NSV and GV, with NSV showing a quadratic response, reaching yield plateau within the N8–N12 range, and the yield decreased drastically when N rate was raised further (mainly due to lodging), whereas the yield of GV displayed a linear response to N rate (thanks to their lodging resistance, Figure 1a,b). For GV, the yield in N20 was the highest (Tables 3 and S1).

**Table 3.** Effects of different nitrogen reduction treatments on the yield and its components of nitrogen-saving rice varieties and general nitrogen rice varieties (2021).

| Type [1] | Variety [2] | Nitrogen Treatment | Panicles (×10⁴ ha⁻¹) | Spikelets Per Panicle | Seed Setting Rate (%) | 1000-Grain Weight (g) | Yield (t·ha⁻¹) |
|---|---|---|---|---|---|---|---|
| NSV | WMJ | N0 | 248.83 ± 1.91 c | 125.23 ± 0.76 d | 89.37 ± 0.23 a | 28.09 ± 0.03 a | 7.82 ± 0.07 d |
| | | N8 | 308.20 ± 2.98 a | 148.17 ± 0.40 b | 85.19 ± 0.37 bc | 27.66 ± 0.07 b | 10.76 ± 0.07 b |
| | | N12 | 308.91 ± 3.69 a | 152.36 ± 0.94 a | 85.64 ± 0.17 b | 27.40 ± 0.09 b | 11.04 ± 0.10 a |
| | | N16 | 281.26 ± 0.66 b | 152.26 ± 0.54 a | 84.51 ± 0.18 c | 27.46 ± 0.13 b | 9.94 ± 0.07 c |
| | | N20 | 287.69 ± 3.24 b | 142.37 ± 0.81 c | 81.72 ± 0.45 d | 27.49 ± 0.19 b | 9.84 ± 0.13 c |
| | YNJ | N0 | 250.51 ± 2.11 d | 127.56 ± 0.40 d | 89.80 ± 0.45 a | 28.08 ± 0.07 a | 8.06 ± 0.02 e |
| | | N8 | 308.88 ± 2.03 a | 151.05 ± 0.77 b | 85.48 ± 0.08 b | 27.73 ± 0.06 ab | 11.06 ± 0.12 b |
| | | N12 | 311.98 ± 1.00 a | 155.06 ± 0.95 a | 85.59 ± 0.09 b | 27.48 ± 0.09 b | 11.38 ± 0.08 a |
| | | N16 | 287.61 ± 1.63 c | 154.86 ± 0.55 a | 84.43 ± 0.18 c | 27.53 ± 0.12 b | 10.35 ± 0.08 c |
| | | N20 | 297.85 ± 3.06 b | 144.80 ± 0.82 c | 81.84 ± 0.41 d | 27.57 ± 0.18 b | 9.73 ± 0.11 d |
| GV | HD5 | N0 | 221.87 ± 6.72 c | 112.21 ± 2.18 d | 92.53 ± 0.37 a | 28.55 ± 0.04 a | 6.57 ± 0.07 d |
| | | N8 | 271.10 ± 3.96 b | 130.94 ± 0.59 c | 90.53 ± 0.16 b | 26.75 ± 0.11 c | 8.60 ± 0.15 c |
| | | N12 | 285.15 ± 3.60 a | 134.64 ± 0.77 b | 86.83 ± 0.40 c | 27.17 ± 0.21 b | 9.06 ± 0.02 b |
| | | N16 | 291.24 ± 3.02 a | 137.13 ± 0.74 a | 85.58 ± 0.12 d | 27.39 ± 0.25 b | 9.36 ± 0.07 a |
| | | N20 | 288.49 ± 5.39 a | 137.50 ± 0.44 a | 87.58 ± 0.73 c | 27.39 ± 0.25 b | 9.51 ± 0.02 a |
| | MGJ200 | N0 | 217.53 ± 1.14 e | 117.75 ± 2.25 d | 92.62 ± 0.37 a | 28.63 ± 0.04 a | 6.79 ± 0.17 e |
| | | N8 | 264.77 ± 3.40 d | 137.49 ± 0.62 c | 90.62 ± 0.16 b | 26.83 ± 0.11 c | 8.85 ± 0.10 d |
| | | N12 | 278.13 ± 6.95 c | 141.62 ± 0.46 b | 86.88 ± 0.45 c | 27.44 ± 0.42 b | 9.39 ± 0.17 c |
| | | N16 | 288.93 ± 2.01 b | 143.98 ± 0.78 a | 85.30 ± 0.57 d | 27.47 ± 0.25 b | 9.75 ± 0.06 b |
| | | N20 | 298.44 ± 2.33 a | 144.37 ± 0.46 a | 87.60 ± 0.73 c | 27.34 ± 0.08 b | 10.32 ± 0.15 a |
| Source of variation | | | | | | | |
| | T | | ** | ** | ** | ns | ** |
| | V | | ns | ** | ns | ns | ** |
| | T × V | | ** | ** | ns | ns | * |
| | N | | ** | ** | ** | ** | ** |
| | T × N | | ** | ** | ** | ** | ** |
| | V × N | | ** | ns | ns | ns | ns |
| | T × V × N | | ns | ns | ns | ns | ** |

[1] NSV, Nitrogen saving rice varieties; GV, General nitrogen rice varieties. [2] WMJ, Wumijing; YNJ, Yangnongjing; HD5, Huaidao5; MGJ200, Mugengjing200. Different letters indicate statistical significance at the *p* = 0.05 level with the same variety in the same column. ns: Not significant at *p* = 0.05 level. * and **: Significant at *p* = 0.05 and 0.01 level, respectively.

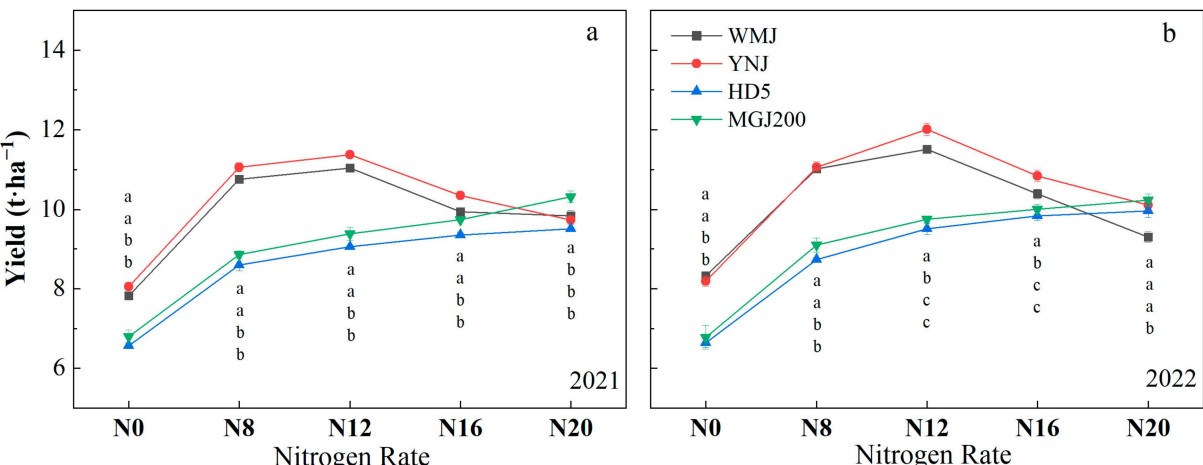

**Figure 1.** The relationship between the rice yield and N rate in 2021 (**a**) and 2022 (**b**). The different color dots represent the ± standard error of the mean. Different letters above or below the color dots indicate statistical significance (*p* < 0.05) within the same treatment. WMJ, Wumijing; YNJ, Yangnongjing; HD5, Huaidao5; MGJ200, Mugengjing200.

Path analysis revealed that the panicle number consistently contributed the most to the yield in all four varieties over the two seasons, followed by spikelets per panicle (Tables 4 and S2), reflecting that the managements in the experiment mainly affected the panicle number and spikelets per panicle.

**Table 4.** Correlation and path analysis of yield and its components (2021).

| Varieties [1] | Yield Components | Direct Path Coefficient | Indirect Path Coefficient | | | | Correlation |
|---|---|---|---|---|---|---|---|
| | | | Panicles | Spikelets per Panicle | Seed Setting Rate | Grain Weight | |
| WMJ | Panicles | 0.7254 | | 0.6040 | −0.3850 | −0.5065 | 0.9830 ** |
| | Spikelets per panicle | 0.3117 | 0.2595 | | −0.1775 | −0.2609 | 0.9150 ** |
| | Seed setting rate | 0.0234 | −0.0124 | −0.0133 | | 0.0158 | −0.5494 * |
| | Grain weight | −0.0152 | 0.0106 | 0.0127 | −0.0102 | | −0.7668 ** |
| YNJ | Panicles | 0.6763 | | 0.5838 | −0.4546 | −0.5069 | 0.9424 ** |
| | Spikelets per panicle | 0.5862 | 0.5060 | | −0.3676 | −0.4711 | 0.9387 ** |
| | Seed setting rate | 0.2765 | −0.1859 | −0.1734 | | 0.1979 | −0.4941 |
| | Grain weight | 0.0720 | −0.0540 | −0.0579 | 0.0516 | | −0.7081 ** |
| HD5 | Panicles | 0.7648 | | 0.7380 | −0.6790 | −0.5872 | 0.9853 ** |
| | Spikelets per panicle | 0.5748 | 0.5546 | | −0.4882 | −0.4366 | 0.9886 ** |
| | Seed setting rate | 0.2210 | −0.1962 | −0.1877 | | 0.1060 | −0.8600 ** |
| | Grain weight | 0.1796 | −0.1379 | −0.1364 | 0.0861 | | −0.7382 ** |
| MGJ200 | Panicles | 0.8297 | | 0.8048 | −0.7046 | −0.5920 | 0.9921 ** |
| | Spikelets per panicle | 0.5126 | 0.4972 | | −0.4340 | −0.3910 | 0.9742 ** |
| | Seed setting rate | 0.2415 | −0.2051 | −0.2044 | | 0.1091 | −0.8150 ** |
| | Grain weight | 0.1818 | −0.1297 | −0.1387 | 0.0821 | | −0.6921 ** |

[1] WMJ, Wumijing; YNJ, Yangnongjing; HD5, Huaidao5; MGJ200, Mugengjing200. * and **: Significant at $p = 0.05$ and 0.01 level, respectively.

*3.2. Tillering and Productive Tiller Percentage (PTP)*

Within 35 days after transplanting, the tiller number increased drastically and reached a maximum number, irrespective of the varieties. However, tillering in the NSV was apparently faster than that in the GV, especially at the initial three weeks after transplanting ($p < 0.05$) (Figures 2 and S1). Interestingly, the differences in the number of stems and tillers between N8 and N0 were bigger in the NSV than the GV, and their responses to further increasing in N rate (N12–N20) displayed an opposite trend (Figures 2 and S1). These indicated that the tillering in the NSV was more responsive to lower N input than the GV, though the maximum number of tillers was similar in all of them.

The productive tiller percentages (PTPs) of the GV (HD5 and MGJ200) were significantly higher than those of the NSVs (WMJ and YNJ), especially under the higher end of the N rate (N16 and N20) (Figure S2). However, when the N rate was reduced to N12 or even lower, the PTPs of the NSVs were generally comparable to those of the GVs, though the difference was not significant. These indicated that adopting NSVs could not improve the PTPs, even under a reduced N rate.

*3.3. N Absorption and Use Efficiency*

For the grain production efficiency of N uptake (NUEg, Figure 3a,b), the highest numbers were all achieved at blank control (N0). Along with an increased N rate, NUEg decreased significantly in all four varieties ($p < 0.05$). However, the NUEg dropped less in the NSVs (WMJ and YNJ) than in the GVs (HD5 and MGJ200), especially at the N8 level (compared to N0). Interestingly, when the N rate increased from N8 to N20, the NUEg decreased more drastically in the NSVs than in the GVs. Similar contrasting responses were observed in the agronomic N use efficiency (NAE, Figure S3a,b), N partial factor productivity (NPFP, Figure S3c,d), and N physiological efficiency (NPE, Figure S3e,f) between the NSVs and GVs, though the flip point of the N rate was different. These results

indicated that the general NUE for the NSVs was higher than the GVs, especially under limited N input conditions.

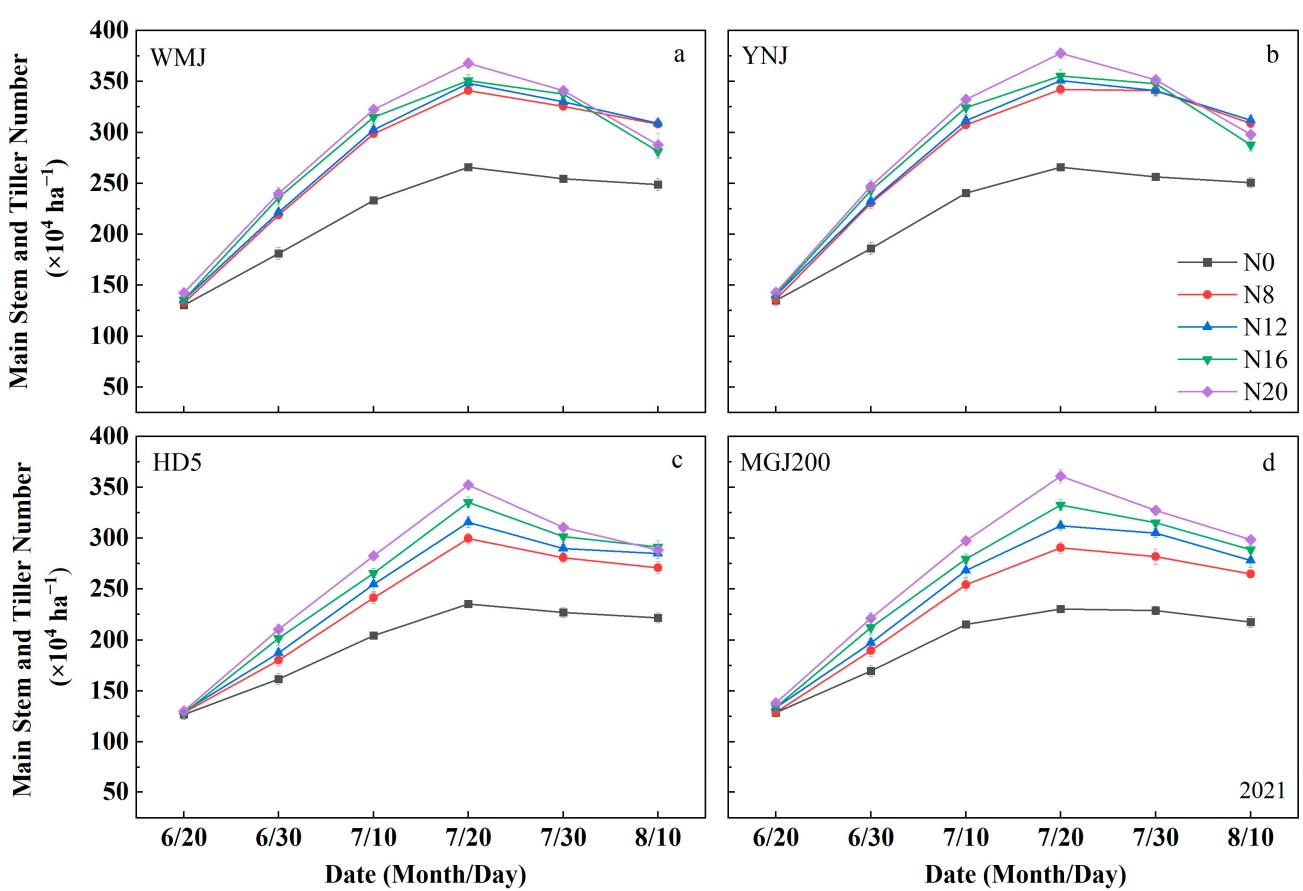

**Figure 2.** The relationship between the main stem and tiller number of varieties WMJ (**a**), YNJ (**b**), HD5 (**c**), MGJ200 (**d**) in the growth period of 2021. The different color dots represent the ± standard error of the mean. WMJ, Wumijing; YNJ, Yangnongjing; HD5, Huaidao5; MGJ200, Mugengjing200.

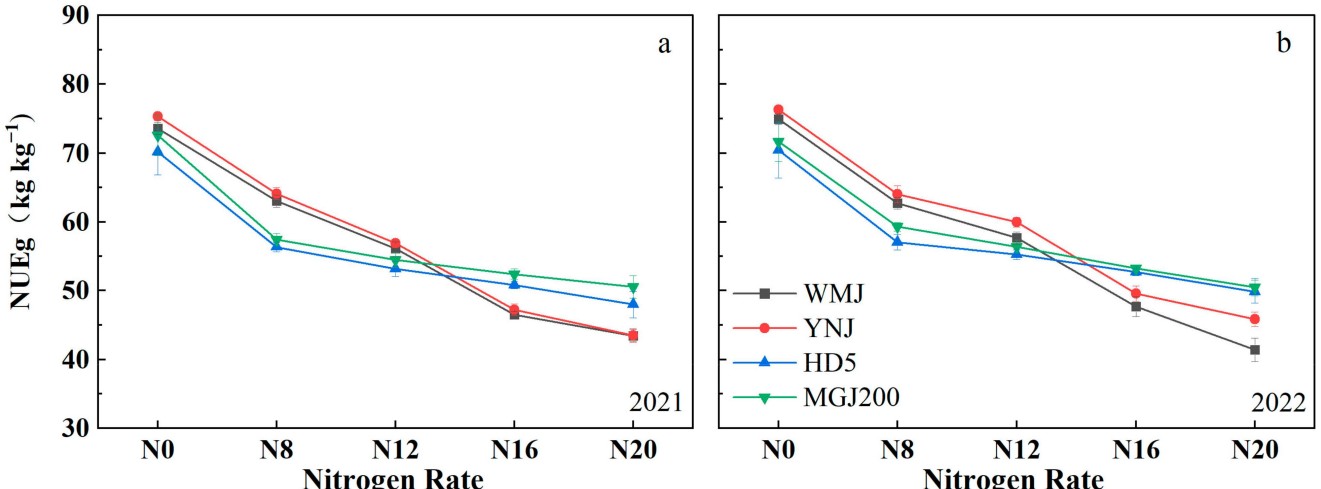

**Figure 3.** The relationship between the NUEg and N rate in 2021 (**a**) and 2022 (**b**). The different color dots represent the ± standard error of the mean. NUEg; nitrogen use efficiency for grain; WMJ, Wumijing; YNJ, Yangnongjing; HD5, Huaidao5; MGJ200, Mugengjing200.

When plotting the yield against the N accumulation at maturity stage, a quadratic regression was observed in the NSVs, while a linear regression was revealed in the GVs (Figure 4). These results suggested that the characteristics of N accumulation and yield formation were very different in the two types of varieties.

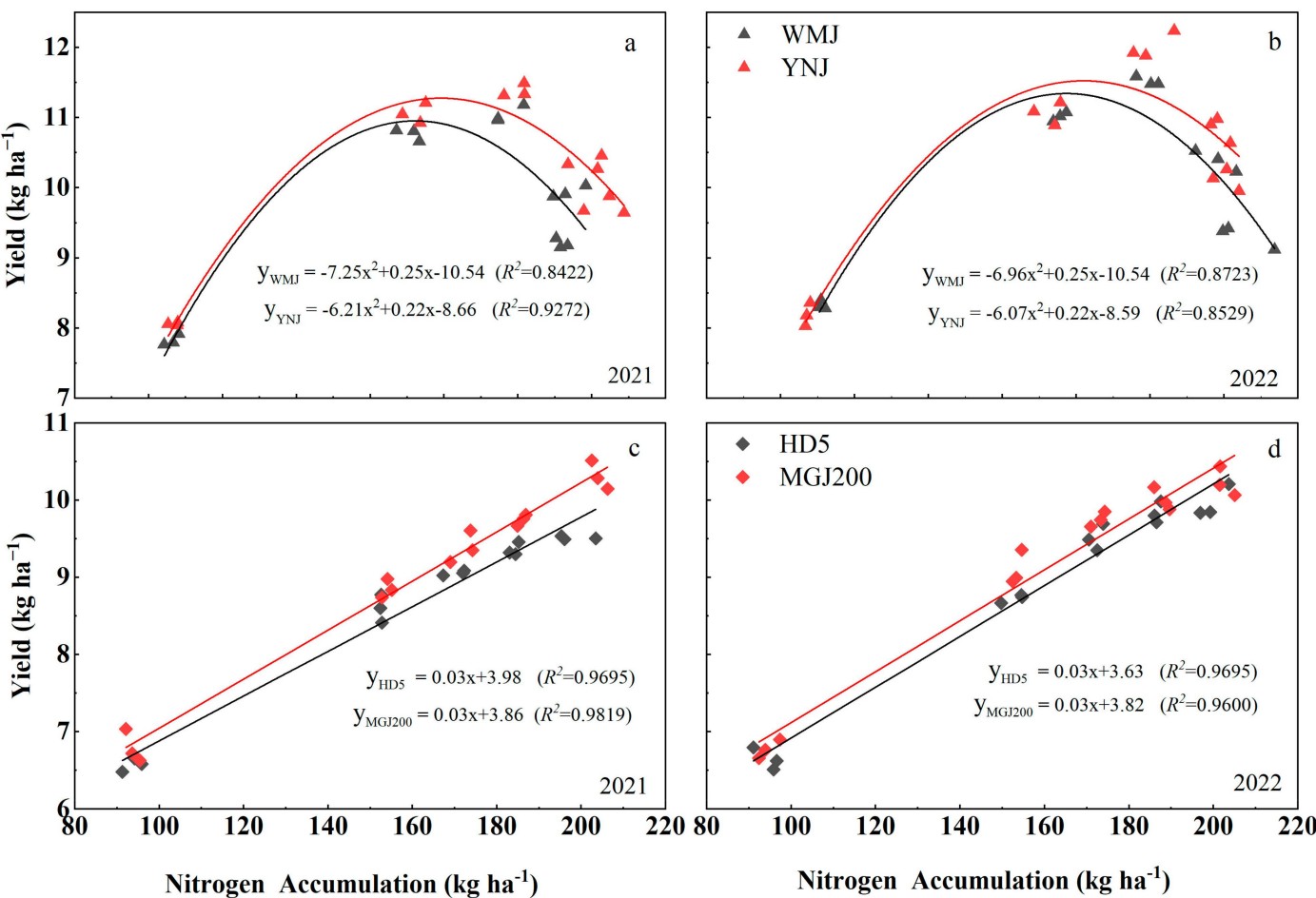

**Figure 4.** The relationship between the yield of NSV (**a,b**), GV (**c,d**) and N accumulation in two years. NSV, N saving varieties; GV, general N varieties. WMJ, Wumijing; YNJ, Yangnongjing; HD5, Huaidao5; MGJ200, Mugengjian200.

*3.4. Non-Structural Carbohydrate (NSC) Remobilization*

NSC may contribute up to 30% of the yield in *japonica* rice varieties [21]. Our results showed that the NSC was similar in the two types of varieties at N0, but a significantly greater accumulation of NSC at heading and maturity, re-mobilization, and contribution of NSC to grain yield was observed in N8 and N12 in the NSVs (Figure 5). However, at the highest N rate (N20), the GVs displayed a higher contribution of accumulation, re-mobilization, and contribution of NSC than the NSVs.

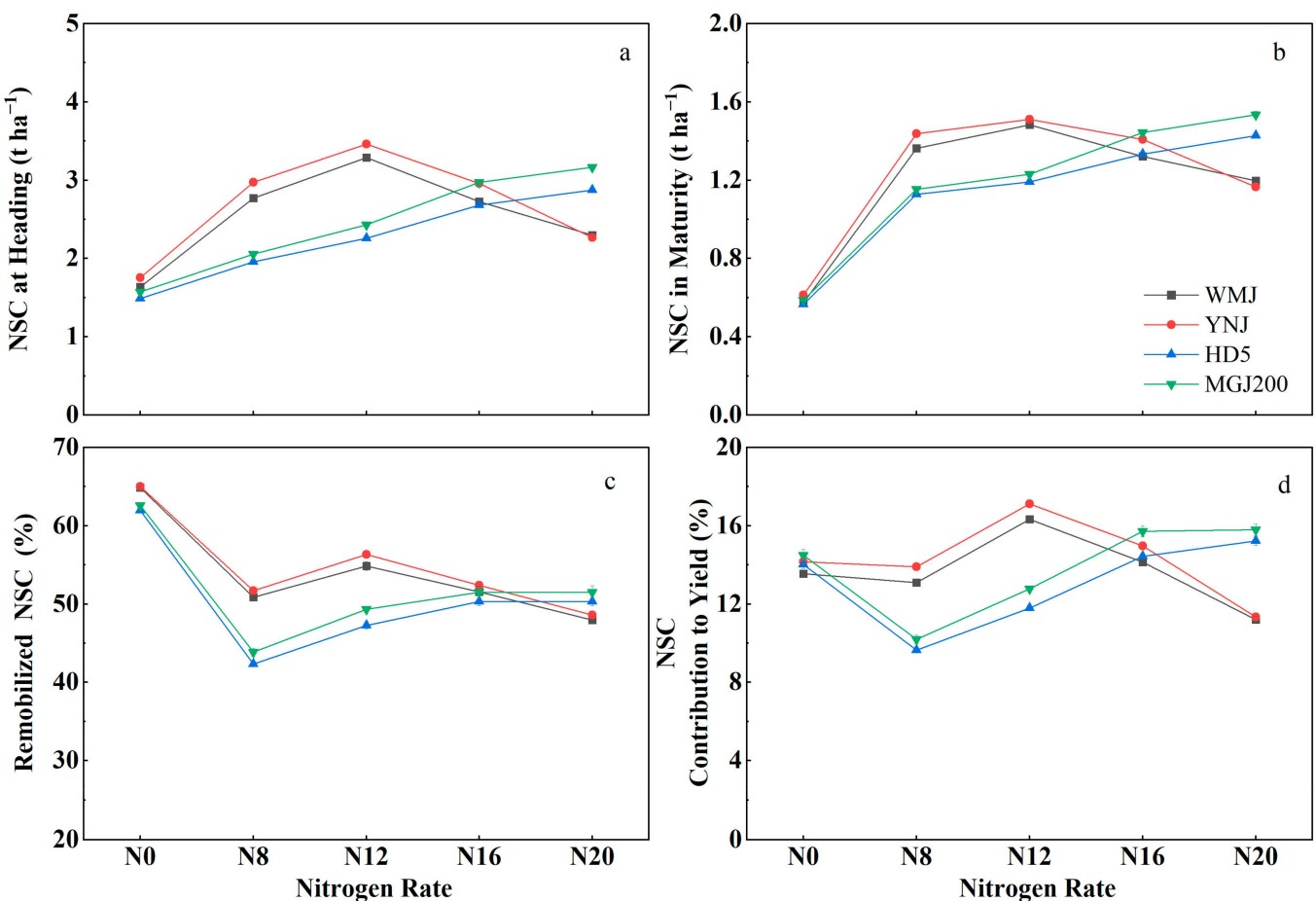

**Figure 5.** The relationship between the NSC at heading (**a**), NSC in maturity (**b**), remobilized NSC (**c**), NSC contribution to yield (**d**) and N rate. The different color dots represent the ± standard error of the mean. NSC, non-structural carbohydrate. WMJ, Wumijing; YNJ, Yangnongjing; HD5, Huaidao5; MGJ200, Mugengjing200.

Under N8 and N12, NSC re-mobilization in WMJ and YNJ was 50.85% and 51.69%, and 54.88% and 56.32%, respectively (Figure 5c). Their contribution to grain yield in WMJ and YNJ was 13.10% and 13.91%, and 16.33% and 17.13%, respectively (Figure 5d). The majority of the NSC was re-mobilized, and it played a relatively minor role in the yield formation. A plausible assumption is that the major contribution was directly from photosynthetic products.

*3.5. Correlation of NSVs and GVs with Yield and N Use Efficiency*

In this study, rice yield, panicles, spikelets per panicle (SP), and seed setting rate (SSR) of NSVs were positively correlated with NUEg, NAE, NPFP, and NPE (Figure 6a) and negatively correlated with those of GVs (Figure 6b). Yield, panicles, and SSR of NSV were negatively correlated with the N accumulation. However, yield, panicles, and SSR of GVs were positively correlated with the N accumulation.

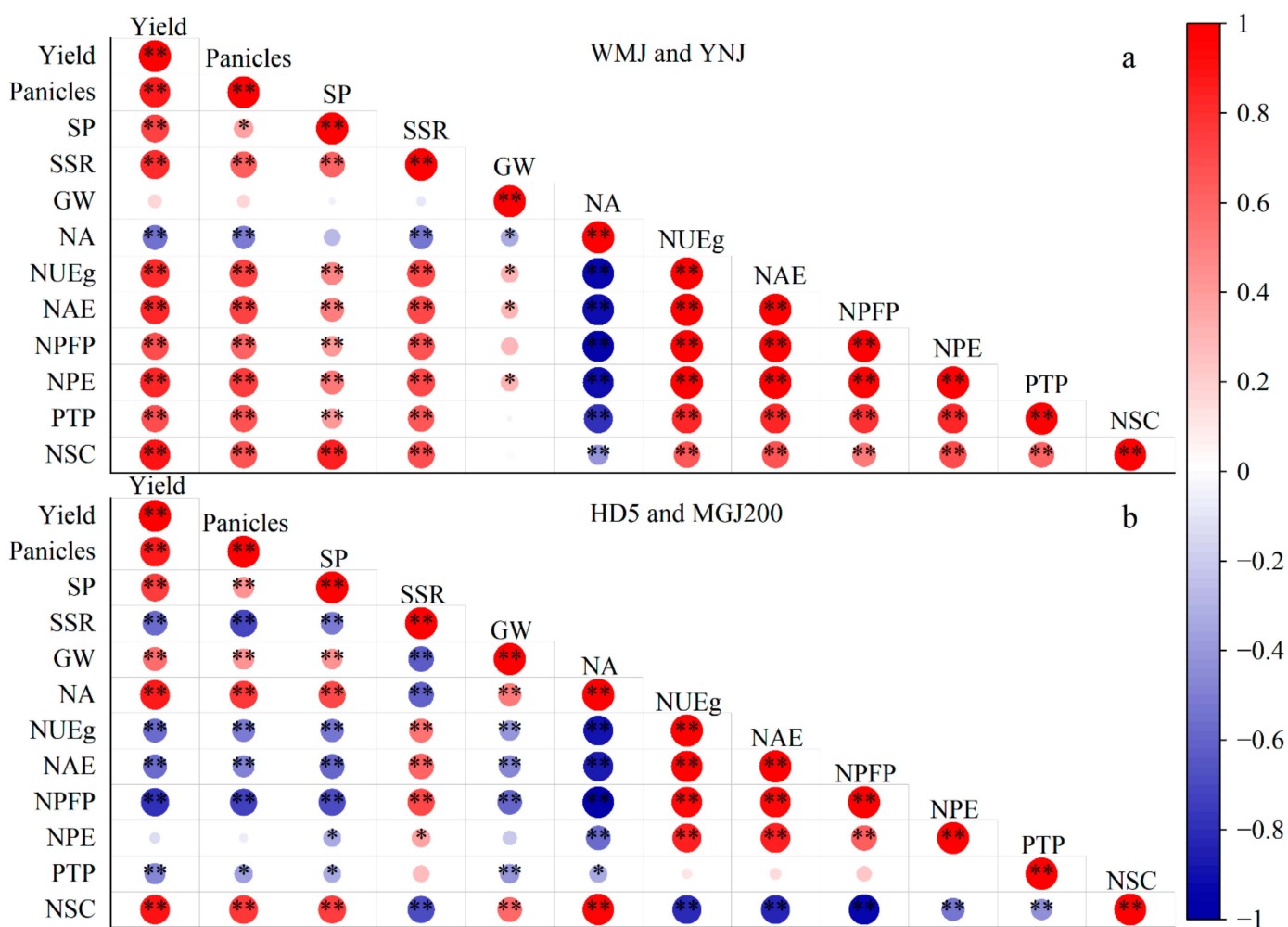

**Figure 6.** The correlation of NSV (**a**) and GV (**b**) with yield and N use efficiency. * and **: Significant at *p* = 0.05 and 0.01 level, respectively. SP, spikelets per panicle; SSR, seed setting rate; GW, 1000-grain weight; NA, nitrogen accumulation; NUEg; nitrogen use efficiency for grain; NAE, agronomic nitrogen use efficiency; NPFP, nitrogen partial factor productivity; PTP, productive tiller percentage; NSC, non-structural carbohydrate. WMJ, Wumijing; YNJ, Yangnongjing; HD5, Huaidao5; MGJ200, Mugengjing200.

## 4. Discussion

N fertilization is an important management regulation in rice growth and yield formation [22,23]. Limited acreage availability makes maintaining or raising the yield per unit area the only reliable approach to secure grain yield output [10,24]. Because of the false belief in its magical effect, very often, excessive N fertilizer input accompanies unstable yield, low N use efficiency, a hike in cost, and aggravated environmental pollution [25,26]. Therefore, it is critical to optimize the proper input of N and to improve the N use efficiency of crops [27–29]. Breeding and adoption of the N-saving variety (NSVs or high N use efficiency variety) are likely the ultimate solution to this issue. However, limited options of NSVs are available, and reports on their responses to a reduced N level are rare.

Many reports have revealed the N optimization management of the Taihu Lake region (200 Km south of Yangzhou) with an N rate of 270–360 Kg ha$^{-1}$ and yield output 9–13 t ha$^{-1}$ in GV *japonica* varieties [29,30]. In this study, the maximum N uptake of NSV was obtained at N12 (180 Kg N ha$^{-1}$), while the N uptake reduced significantly at N20 (300 Kg N ha$^{-1}$). The NUEg, NAE, NFPF, and NPE of NSVs were all higher than those of GVs under the N reduction level. On the contrary, the NUE, NPE, and NFPF of conventional

varieties were higher than those NSVs in the local common N rate range (300 Kg N ha$^{-1}$) and maintaining average grain yield.

The results here showed that the two newly released NSV varieties WMJ and YNJ from Yangzhou University displayed higher N use efficiency at lower N input (N8–N12, equal to 120–180 Kg N ha$^{-1}$). Compared to the local average N input (N20, 300 Kg N ha$^{-1}$) and rice grain yield output of 9–10 t ha$^{-1}$, the NSVs could achieve an acceptable target grain yield of 10.7–12.5 t ha$^{-1}$ at an N rate of 120–180 Kg N ha$^{-1}$. This is a reduction of 40–60% of its common local N fertilizer input, without any sacrifice in grain yield. It is speculated that the NSVs may possess N efficient utilization genes, and a follow-up should be undertaken on how and why they respond to lower N input in a favorable pattern. Should this practice be widely adopted, the economic, environmental, and social gains would be very impressive for sustainable agriculture.

Unfortunately, nothing good comes without a bad side. These two NSVs were more susceptible to lodging, especially when the N rate was over 240 Kg ha$^{-1}$. This vulnerability to lodging was specifically real for YNJ. Though we did not present data on the percentage of lodging, it happened in all experimental plots in N16 (240 Kg ha$^{-1}$) and N20 (300 Kg ha$^{-1}$) in both years. Should these NSVs be improved in lodging resistance, it would be of greater value in sustainable rice production. Even in their current shape, these two NSVs are good choices when strictly applying an N rate lower than 120–150 Kg ha$^{-1}$ for a medium soil fertility. Lodging is related to the remobilization of NSC after flowering; a high remobilization percentage of NSC in the NSVs may contribute to lodging vulnerability in these NSVs.

High productive tiller percentage (PTP) is a healthy index for population quality [31,32], and rice grain yield is associated with tillering capacity and panicle number under low-N input [22,33]. Our results corroborated these findings, where high PTP was achieved under a reduced N level for the NSVs and GVs, and the grain yield was in an acceptable range for the NSVs. However, when excessive N was applied, the PTP showed a drastic reduction in the NSVs while the GVs lowered gradually, indicating that PTP is still a useful benchmark for monitoring field population quality.

## 5. Conclusions

Reducing the N application could not only maintain the yield in NSVs (WMJ and YNJ) but also raise the NUE. At an N rate 180 Kg ha$^{-1}$, the NSVs excelled in tillering number, panicles, higher total spikelet number, PTP, biomass accumulation, NUEg, NAE, NPFP, NPE, and NSC remobilization, resulting in a balanced grain yield and N efficiency. Lower N input is more suitable for NSVs than GVs in the lower reaches of the Yangtze River in China. We propose to reduce N fertilizer to a range of N8-N12 for NSV in sustainable agriculture practices.

**Supplementary Materials:** The following supporting information can be downloaded at: https://www.mdpi.com/article/10.3390/agronomy13071832/s1, Table S1: Effects of different nitrogen reduction treatments on the yield and its components of nitrogen-saving rice varieties and general nitrogen rice varieties (2022); Table S2: Correlation and path analysis of yield and its components (2022); Figure S1: The relationship between the main stem and tiller number of varieties WMJ (a), YNJ (b), HD5 (c), MGJ200 (d) in the growth period of 2022.; Figure S2: The relationship between the productive tiller percentage and nitrogen rate in 2021 (a) and 2022 (b). Figure S3: The relationship between the NAE (a, b), NPFP (c, d), NPE (e, f) and N rate in two years. The different color dots represent the ± standard error of the mean. NAE, agronomic nitrogen use efficiency; NPFP, nitrogen partial factor productivity; WMJ, Wumijing; YNJ, Yangnongjing; HD5, Huaidao5; MGJ200, Mugengjing200.

**Author Contributions:** Conceptualization, X.Z. and Y.Y.; methodology, X.Z. and Y.Y.; validation, H.Z.; formal analysis, X.Z. and X.S.; investigation, Z.W. and Y.C.; resources, J.Z., N.X. and C.P.; data curation, Y.G., X.L., N.H. and C.J.; writing—original draft preparation, X.Z.; writing—review and editing, Y.Y.; visualization, G.D.; supervision, A.L. and Y.Y.; project administration, J.H.; funding acquisition, X.Z. and Y.Y. All authors have read and agreed to the published version of the manuscript.

**Funding:** This research was funded by the Natural Science Foundation of Jiangsu (BK20201219), the National Natural Science Foundation of China (31571608), and Natural Science Fund for Colleges and Universities in Jiangsu Province (15KJA210003), Key R&D Program of Jiangsu Province Modern Agriculture (BE2022335), Yangzhou Modern Agriculture Project (YZ2022043).

**Data Availability Statement:** The data presented in this study are available upon request from the corresponding author.

**Acknowledgments:** We are grateful to Shuzhu Tang for providing the venues and meaningful help.

**Conflicts of Interest:** The authors declare no conflict of interest.

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
