# Peer review of "Reduced Nitrogen Input Combined with Nitrogen-Saving japonica Rice Varieties Balances Yield and Nitrogen Use Efficiency in The Lower Reaches of the Yangtze River in China"

_agronomy, doi:10.3390/agronomy13071832_

Round 1
Reviewer 1 Report
The work presented to review is very interesting and contains important information regarding cultivation of rice , nitrogen fertilization and environmental protection. However, I have two remarks:
- in the conclusions, it is worth adding information which variety responded most favorably,
- the 2-year study should be treated as preliminary and continue research to draw reliable conclusions.
Author Response
Dear Reviewer,
Thank you very much for these examinations on our manuscript entitled “Reduced Nitrogen Input Combined with Nitrogen-Saving Japonica Rice Varieties Balances Yield and Nitrogen Use Efficiency in The Lower Reaches of the Yangtze River in China”. These valuable comments have greatly improved the quality and value of this manuscript. Here, we revised the manuscript in detail according to your comments, and point-by-point responses to your comments are listed below.
Point 1. In the conclusions, it is worth adding information which variety responded most favorably.
Response 1: Thank you for the comment. We have added the information as follow line 349 and lines 352-353 in the revised manuscript.
Point 2. The 2-year study should be treated as preliminary and continue research to draw reliable conclusions.
Response 2: Thank you for your comment. We have added the reliable conclusions as follow lines 327-329 in the revised manuscript.
Once again, we thank you for your careful assessment of our manuscript and the valuable comments provided.
We thank you again and look forward to hearing from you.
Sincerely,
Xiaogxiang Zhang

Reviewer 2 Report
Background:
Maintaining rice yield and reducing nitrogen (N) input are two important target birds hard to shot at one stone. Adoption of nitrogen saving variety (NSV) provides an unique opportunity to achieve it. However, limited options in NSV japonica rice and lack information on their responses to N reduction make it difficult to management decisions. This study aims to explore the responses of yield and nitrogen use efficiency (NUE) in NSV to N reduction. Two newly released NSV and two popular general varieties (GV) of japonica rice were field test in Yangzhou, located at the lower reaches of Yangtze River of China, in two consecutive years. The results showed that for NSV, a 40-60% reduction of common practice N rate (300 Kg ha-1), the rice yield could maintain a record average level, whereas the yield for the GV would drop 20-30%. This indicates that combining practices of adoption of NSV and N reduction to 120-180 Kg N ha-1 could balance the yield and N consumption. Moderate N reduction promotes the N accumulation and NUE, increases the number of tillers, the productive tiller percentage and the total amount of spikelets in the population, and increases the carbon and N metabolism of the population in the NSV. Compared with GV, NSV showed higher NUE and non-structural carbohydrate re-mobilization in the reduced N rate. We conclude that practice of N reduction has to adopt NSV at the same time in order to maintain grain yield level in rice.
General comment: This is an important topic relevant to the journal and to the readership. The methodology is sound, the results are generalizable, and add significantly to what is already known about this topic. There are no problems with ethics or conflict of interest."
1. there is plenty of similar studies related to this study, nitrogen fertilizers, NUE, etc. (please see below that you can consult with their reference lists), that need to be more connected with the specific problem of this study.
10.2174/1874331502014010246
10.1080/00103624.2021.1885687
1-Do the authors need to explain why they choose the selected topic?
Abstract:
1. Write the importance of this study in the first sentence with references…
2. The statistical results should be included in the abstract.
3. Please delete “We” “Our” words from abstract and introduction….
Introduction:
Overall, the writing is impressive and the introduction covers a range of studies related to NUE. However, it would be beneficial to establish a stronger connection between these studies and the specific problem addressed in this research. I suggest consulting the reference lists of these studies for additional sources that could be used to strengthen the discussion of the research problem.
1. Update all the old references (published before 2016) with recent references….
2. Write the novelty and the main objectives of this study….
3. The introduction section is short. Make it sharp……
Methodology:
very well written.
1. Do the authors need to explain why they choose Rice as a test plant?
2. What methods were used to measure the changes in chemical composition and functional groups in the N fertilizers?
3. Could you provide more information on the specific changes observed in the soil microbial community structure?
Results
very well written.
1. The figures are too much (7 Tables)??
2. Results part should be sharp not lengthy.
3. Please describe the significant differences according to least significant difference at p<0.05.”
Discussion
1. The scientific discussion should be strengthen. Expand the scientific reason/mechanism behind the results….
Conclusion:
Please strength the conclusion is very short and incorporate a discussion of the study's limitations and future directions within the same section to address them.
References:
1. The References are too much (40 references)??
" Update all the old references (published before 2016) with recent references."
Thank you
Author Response
Dear Reviewer,
Thank you very much for these examinations on our manuscript entitled “Reduced Nitrogen Input Combined with Nitrogen-Saving Japonica Rice Varieties Balances Yield and Nitrogen Use Efficiency in The Lower Reaches of the Yangtze River in China”. These valuable comments have greatly improved the quality and value of this manuscript. Here, we revised the manuscript in detail according to your comments, and point-by-point responses to your comments are listed below.
Point 1. General comment: Do the authors need to explain why they choose the selected topic?
Response 1: Thank you for your comment. We have added the information as follow lines 67-68 in the revised manuscript.
Point 2. Abstract: Write the importance of this study in the first sentence with references.
Response 2: Thank you for the comment. We have modified the first sentence as follow lines 17-18 in the revised manuscript.
Point 3. Abstract: The statistical results should be included in the abstract.
Response 3: Thank you for the valuable comment. We have added the statistical results in the revised manuscript.
Point 4. Abstract: Please delete “We”, “Our” words from abstract and introduction.
Response 4: Thank you for the valuable comment. We have made the necessary revisions in accordance with your comments as follow line 31, lines 64-65 and line 70 in the revised manuscript.
Point 5. Introduction: Update all the old references (published before 2016) with recent references.
Response 5: Thank you for the valuable comment. We had updated all the old references (published before 2016) with recent references in the revised manuscript.
Point 6. Introduction: Write the novelty and the main objectives of this study.
Response 6: Thank you for your comment. We have made the necessary revisions as lines 65-67 lines in the revised manuscript.
Point 7. Introduction: The introduction section is short. Make it sharp.
Response 7: Thank you for the comment. We have made necessary additions in the revised manuscript.
Point 8. Methodology: Do the authors need to explain why they choose Rice as a test plant?
Response 8: Thank you for the valuable comment. We have made necessary additions as lines 88-89 and lines 92-94 in the revised manuscript.
Point 9. Methodology: What methods were used to measure the changes in chemical composition and functional groups in the N fertilizers?
Response 9: Thank you for your comment. The methods were listed as follow lines 135-136 in the revised manuscript.
Point 10. Methodology: Could you provide more information on the specific changes observed in the soil microbial community structure?
Response 10: Thank you for the valuable comment. At present, we are mainly concerned about nitrogen-saving varieties, and in the following studies, we will consider changes in soil microbial community structure.
Point 11. Results: The figures are too much (7 Tables)??
Response 11: Thank you for your comment. We have adjusted the number of Figures in the revised manuscript.
Point 12. Results: Results part should be sharp not lengthy.
Response 12: Thank you for your comment. We have made necessary changes to results in the revised manuscript.
Point 13. Results: Please describe the significant differences according to least significant difference at p<0.05.”
Response 13: Thank you for the comment. We have added the significant differences in the revised manuscript.
Point 14. Discussion: The scientific discussion should be strengthen. Expand the scientific reason/mechanism behind the results.
Response 14: Thank you for the comment. We have expanded the scientific reason as following lines 327-329 in the revised manuscript.
Point 15. Conclusion: Please strength the conclusion is very short and incorporate a discussion of the study's limitations and future directions within the same section to address them.
Response 15: Thank you for your comment. We have made necessary changes to the conclusion in the revised manuscript.
Point 16. References: The References are too much (40 references). " Update all the old references (published before 2016) with recent references."
Response 16: Thank you for the comment. We have deleted the old references (published before 2016) and updated some recent references in the revised manuscript.
Once again, we thank you for your careful assessment of our manuscript and the valuable comments provided.
We thank you again and look forward to hearing from you.
Sincerely,
Xiaogxiang Zhang